# Research Advances in Nucleic Acid Delivery System for Rheumatoid Arthritis Therapy

**DOI:** 10.3390/pharmaceutics15041237

**Published:** 2023-04-13

**Authors:** Xintong Zhang, Yanhong Liu, Congcong Xiao, Youyan Guan, Zhonggao Gao, Wei Huang

**Affiliations:** 1State Key Laboratory of Bioactive Substance and Function of Natural Medicines, Institute of Materia Medica, Chinese Academy of Medical Sciences and Peking Union Medical College, Beijing 100050, China; grace@imm.ac.cn (X.Z.); liuyanhong@imm.ac.cn (Y.L.); xiaocongcong@imm.ac.cn (C.X.); 2Beijing Key Laboratory of Drug Delivery Technology and Novel Formulations, Department of Pharmaceutics, Institute of Materia Medica, Chinese Academy of Medical Sciences and Peking Union Medical College, Beijing 100050, China; 3Department of Urology, National Cancer Center/National Clinical Research Center for Cancer/Cancer Hospital, Chinese Academy of Medical Sciences and Peking Union Medical College, Beijing 100021, China; guanyouyan@cicams.ac.cn

**Keywords:** rheumatoid arthritis, nucleic acid delivery system, gene therapy, non-viral vectors, nanomaterials

## Abstract

Rheumatoid arthritis (RA) is a chronic inflammatory autoimmune disease that affects the lives of nearly 1% of the total population worldwide. With the understanding of RA, more and more therapeutic drugs have been developed. However, lots of them possess severe side effects, and gene therapy may be a potential method for RA treatment. A nanoparticle delivery system is vital for gene therapy, as it can keep the nucleic acids stable and enhance the efficiency of transfection in vivo. With the development of materials science, pharmaceutics and pathology, more novel nanomaterials and intelligent strategies are applied to better and safer gene therapy for RA. In this review, we first summarized the existing nanomaterials and active targeting ligands used for RA gene therapy. Then, we introduced various gene delivery systems for RA treatment, which may enlighten the relevant research in the future.

## 1. Introduction

Rheumatoid arthritis (RA) is a chronic inflammatory autoimmune disease, distinguished by synovial hyperplasia and the destruction of cartilage and subchondral bone, which can result in persistent joint pain, stiffness, swelling and even work disability in severe cases. Apart from articular destruction, RA can also cause systemic complications such as anemia and pericarditis [1,2,3,4]. Nearly 1% of the total population suffers from RA worldwide [5] and the long-time treatment causes a huge economic burden for patients. The exact causes of RA remain unknown. It is believed that the infiltration, rapid proliferation and activation of different kinds of inflammatory cells, such as macrophages, B cells, T cells, dendritic cells and mast cells, as well as the release of invasive proteases such as matrix metalloproteinases and inflammatory cytokines (e.g., IL-1β, IL-6 and TNF-α) mainly contribute to the development of the disease [2,6]. The interaction of multiple signal pathways, such as the NF-κB signaling pathway, JAK/STAT signaling pathway, MAPK signaling pathway, Toll-like receptors signaling pathway and Wnt signaling pathway, is also involved in the occurrence of RA [1,7].

Four kinds of drugs are usually used for the conventional treatment of RA. They are non-steroidal anti-inflammatory drugs (NSAIDs), glucocorticoids, non-biologic disease-modifying anti-rheumatic drugs (DMARDs) and biological agents [8]. However, each of them has some limitations. Generally, the first choice of therapies for RA are NSAIDs, such as celecoxib and piroxicam, which can only relieve the symptoms instead of having any effect on the cause of RA. Furthermore, they can cause some side effects such as peptic ulcer disease, dyspepsia and bleeding. Glucocorticoids such as dexamethasone have great anti-inflammatory and analgesic effects, but they can also cause some adverse effects such as osteoporosis, weight gain and Cushing’s syndrome. Therefore, they need to be used under the guidance of an experienced doctor. DMARDs such as methotrexate (MTX) can influence the progression of RA by preventing the further destruction of the joint. Although they have been found successful in the treatment of RA, they still have some severe side effects, including hepatic cirrhosis and rarely pneumonitis. Biological agents possess high specificity and efficacy, and the representative of this category is adalimumab, which is a kind of monoclonal antibody against TNF-α. However, the side effects, such as reduced responsiveness after multiple administrations and increased risk of infection, can not be ignored [6,9]. As RA is a chronic disease, the safety and cost of long-term therapy need to be taken into consideration, and innovative agents or therapeutic strategies are required to overcome the limitations mentioned above.

Gene therapy is a new technology that introduces specific DNA or RNA into the patient to change gene expression with the aim of treating or preventing diseases [10]. The idea of gene therapy has been around for several decades, but it was only in the 1980s that scientists began to develop the tools and techniques necessary to make it a reality. In the 1990s, there was an acceleration in gene therapy trials, marked by the first authorized attempt to use retroviral vectors expressing adenosine deaminase (ADA) in a gene therapy trial for treating ADA deficiency, which causes severe combined immunodeficiency [11]. Gene therapy possesses many advantages, including treating genetic disorders, achieving more precise treatment of diseases, long-lasting effects and potential for personalized medicine, which provides a new treatment method for some currently difficult-to-treat diseases. With an improved understanding of RA etiology, gene therapy may be a potential means to treat RA, as it can block the production and activity of biological agents causing inflammation or express anti-inflammatory proteins, which can not be delivered directly because of their instability and short half-time in vivo. The gene delivery system is very important to in vivo gene therapy, as the instability of the nucleic acid makes it easy to degrade by the nuclease during circulation. Furthermore, high molecular weight and negative charges prevent it from efficient cellular uptake or lysosomal escape [6]. Therefore, lots of attention has been paid to the development of gene delivery vehicles to take gene therapeutics to the expected sites successfully [12]. With the development of materials science, pharmaceutics and pathology, more novel materials have been synthesized, and some intelligent delivery strategies have been created to achieve more effective and safer gene therapy for RA. However, the existing reviews mainly focus on the potential therapeutic nucleic acids for RA gene therapy, while the introduction of gene delivery systems is still in a blank state. Therefore, this review mainly summarizes the research progress of nucleic acid delivery systems for RA in recent years. We first summarize the innovative nanomaterials and diverse active targeting strategies used for RA gene delivery. Then, the recent development of delivery systems for RA gene therapy is discussed according to different therapeutic gene targets. Compared with similar reviews in recent years, we not only covered the latest research progress in this area but also introduced the delivery systems of other kinds of nucleic acids, such as DNA and miRNA, besides the most commonly discussed siRNA. In addition, the treatment strategies targeting cell-free DNA (cfDNA) were also discussed in this review, providing a new approach for RA gene therapy. We hope that new ideas for gene therapy in RA can be inspired by our review.

## 2. Nanomaterials for RA Gene Therapy

Suitable nanomaterial vehicles are vital for gene therapy as they not only prevent nucleic acids from being degraded by nuclease but can also achieve efficient cellular uptake and lysosomal escape, which is very important to effective gene transfection. Meanwhile, many nanomaterials possess specific ligands, such as amino and carboxyl, which can be modified to construct multifunctional and intelligent delivery systems with better specificity and fewer side effects [13,14,15,16]. In this section, we will introduce some representative nanomaterials and their applications in RA gene therapy. The advantages and limitations of the nanomaterials mentioned below are listed in Table 1.

### 2.1. Polyethylenimine

Polyethylenimine (PEI) has been used to deliver genes for a long time. With the high positive charge density, strong buffering ability and proton sponge effect, PEI can prevent genes from being destructed by nuclease and accomplish lysosomal escape. Furthermore, the amino groups of PEI make it possible for various kinds of surface modifications, which enables the delivery system to be multifunctional or target the sites with specificity. PEI, with high molecular weight, possesses better transfection ability. For example, 25 kDa PEI is considered as the “gold standard” effective transfection reagent [28]. However, the higher the molecular weight, the higher the cytotoxicity of PEI. Therefore, many PEI derivatives have been developed to reduce cytotoxicity without affecting the transfection capacity. Using disulfide linkages to synthesize glutathione-responsive disulfide cross-linked PEIs is a method to decrease cytotoxicity and condense genes effectively [29]. As the concentration of glutathione (GSH) in the extracellular environment is much lower (1–10 μM) than that in the intracellular pool (1–10 mM), the disulfide cross-linked PEIs can stay stable before reaching the cell and release genes only in the cytoplasm due to the thiol-disulfide exchange [30]. Yin et al. designed an in situ hydrogel loaded with disulfide-crosslinked polyethyleneimine (PEI-SS) nanoparticles to co-deliver indomethacin (IND), methotrexate (MTX) and siRNA, down-regulating the expression of MMP-9 for the synergistic treatment of RA. The intra-articular administration of the hydrogel possessed great anti-inflammatory ability and could reverse cartilage disruption [17]. Apart from using PEIs and their derivatives alone, many researchers fabricated hybrid micelles composed of PEIs and other polymers. Wang et al. exploited a hybrid micelle system using two kinds of amphiphilic deblock copolymers, PCL-PEI and PCL-PEG, to encapsulate dexamethasone (Dex) and siRNA targeting NF-κB p65. The hybrid micelle could reduce inflammation of mice with collagen-induced arthritis by blocking NF-κB signaling and regulating the ratio of M1 macrophages to M2 macrophages in the arthritic synovium, which provided a novel approach for treating RA [18]. Due to its high transfection efficiency, pH buffering capacity, good endosomal escape ability and the ability to transfect non-dividing cells, PEI is a promising kind of nanomaterial used for nucleic acid delivery systems. Nevertheless, more effort is needed to improve PEI-based nanoparticles with higher specificity and stability and reduced toxicity.

### 2.2. Lipid-Based Nanoparticles

Cationic lipids are usually used to deliver genes, for they can form lipoplexes with polyanionic nucleic acids by attractive electrostatic interactions [20,31,32]. The lipoplexes generally consist of cationic lipids that are necessary for the encapsulation of nucleic acids and helper lipids used to improve stability and transfection efficiency [33]. There are several types of lipid-based nanoparticles, including liposomes, solid lipid nanoparticles, nanostructured lipid carriers, lipid-drug conjugates and lipid-polymer hybrid nanoparticles. Each possesses its own unique properties and advantages. For example, liposomes can be used to encapsulate hydrophilic and hydrophobic drugs and can be functionalized with targeting moieties to improve their specificity to a particular tissue or cell type, while solid lipid nanoparticles have improved stability and longer circulation time compared with liposomes. Many researchers have paid attention to lipid-based nanoparticles used for RA gene therapy. In order to deliver siRNA silencing tumor necrosis factor α (TNF-α) in CIA mice, Khoury et al. used RPR209120 (2-(3-[Bis-(3-amino-propyl)-amino]-propylamino)-N-ditetradecylcarbamoylme-thyl-acetamide) and DOPE (1,2-dioleoyl-sn-glycerol-3-phosphoethanolamine) to construct a liposome carrier system. The delivery system showed a good ability to decrease the level of TNF-α, both systemically and locally, thereby reducing the incidence and severity of arthritis after systemic administration [19]. There are also some limitations on the application of cationic lipoplexes. Because of their high cationic charge, lipoplexes may bind to serum proteins and are not stable in a serum-containing physiological environment. In addition, the cytotoxicity of cationic lipids can not be ignored. To solve this problem, researchers design lipid-polymer hybrid nanoparticles with better colloidal stability via the combination of a polymeric core and a cationic lipid shell layer [20]. Jansen et al. fabricated lipid-polymer hybrid nanoparticles (LNPs) composed of DOTAP (1,2-dioleoyl-3-trimethylammonium propane) and PLGA (poly (DL-lactic-co-glycolic acid) containing siRNA against TNF-α to reduce the chronic inflammatory conditions of RA. In addition to great TNF-α silencing efficiency, LNPs showed better stability than reference lipidoid-based stable nucleic acid particles (SNALPs) [20]. Moreover, the addition of PLGA may provide the nanoparticles with the ability to release cargo in a sustained way due to the outstanding controllable and alterable degradation profile of PLGA particles [34]. Song et al. designed a lipidoid-polymer hybrid nanoparticle composed of Pluronic F127 and spermidine-based lipidoid (S14) to deliver siRNA against IL-1β efficiently. The carriers could lower the expression of IL-1β and effectively attenuate the inflammatory symptoms of arthritis induced by collagen antibody (Figure 1) [21]. Compared with other kinds of nanomaterials, lipid-based nanoparticles possess many unique advantages, including high delivery efficiency, low immunogenicity, good biocompatibility and versatility. However, the limitations, such as low stability, short half-time and toxicity at high concentrations, can not be ignored. Lipid-based nanoparticles have shown promise in preclinical studies and are currently being used in clinical trials for gene therapy. Researchers are working to optimize their stability, delivery efficiency and safety profile to make them more effective in treating diseases. Advancements in nanotechnology and lipid chemistry are expected to lead to the development of more sophisticated lipid-based nanoparticles that can deliver nucleic acids for RA gene therapy with even greater precision and efficiency.

### 2.3. Chitosan

Chitosan is a potential candidate for gene delivery due to its low toxicity, high cationic charge, outstanding biocompatibility and low immunogenicity [35]. Composed of repeating D-glucosamine and N-acetyl-D-glucosamine units, chitosan is a polysaccharide extracted from the exoskeleton of shellfish. Although it possesses lots of advantages, its clinical application is still limited by some challenges, such as poor water solubility, poor targeting ability, charge deduction under physiological condition and premature release in the cytoplasm [36]. Therefore, various strategies and chitosan derivatives have been developed and applied to gene therapy. For instance, the poor solubility of chitosan can be improved by chemical modification such as PEGylation. Chitosan-based nanoparticles have been widely used for RA gene therapy [36]. Shi et al. synthesized a chitosan derivative containing diethylethylamine (DEAE) to increase the solubility and colloidal stability of chitosan nanoparticles, which is due to DEAE ligands’ hydrophilicity and capacity to keep protonated in the physiological environment. Then, they modified this chitosan derivative with folic acid and used it to deliver siRNA against TNF-α for the treatment of RA. The concentration of TNF-α in target tissue was significantly decreased, and the inflammation of CIA mice was attenuated after the intraperitoneal injection of nanoparticles [22]. Lee et al. constructed a nanocomplex of polymerized siRNA (poly-siRNA) silencing TNF-α with thiolated glycol chitosan (tGC) polymers for RA treatment. Compared with methotrexate (5 mg/kg), inflammation and bone erosion in CIA mice was significantly inhibited (Figure 2) [23]. However, the cationic charge density of chitosan may be decreased by excessive modification, which can influence its ability to condense genes [36]. In order to enlarge the application of chitosan and its derivatives in RA gene therapy, more attention is still needed on the modification and delivery approaches to fully take advantage of the unique characteristics of chitosan.

### 2.4. Others

Apart from the nanomaterials mentioned above, there are some other kinds of delivery systems used for RA gene therapy. Iron oxide nanoparticles (IONPs) have been explored as a potential tool in gene delivery systems due to their unique physical and chemical properties, including magnetic properties, high surface area and imaging capabilities. Duan et al. designed polyethyleneimine-functionalized iron oxide nanoparticles to deliver siRNA against the IL-2/-15 receptor β chain to experimental arthritic joints. In addition to the enhanced permeability and retention effect that the RA pathological tissue possesses, the external magnetic field fixed on the skin of the hind paw of AA rats enhanced the accumulation of nanoparticles in inflamed joint tissues and exhibited better therapeutic effects than the treatment without the addition of a magnet [37]. Some cationic peptides can form nanocomplexes with nucleic acid as well [24,38,39]. Kim et al. combined a short AchR-binding peptide derived from the rabies virus glycoprotein (RVG) with nona-D-arginine residues. Then, they used this cationic peptide to deliver siRNA against TNF-α to macrophages and showed that this delivery system could suppress the expression of TNF-α in macrophages and DCs, which might be a potential method for RA treatment [24,25,26]. Various biomimetic nanoparticles have been fabricated for enhanced RA therapy due to their enhanced targeting, biocompatibility and reduced immunogenicity [27,40]. Li et al. used M2 exosomes derived from M2-type macrophages to co-deliver IL-10 pDNA and betamethasone sodium phosphate, which provide a promising approach to RA treatment through the synergistic effect of gene therapies and chemical drugs (Figure 3) [27].

Nanotechnology has shown promising results in the treatment of RA by providing targeted drug delivery systems that can selectively deliver therapeutic agents to the affected sites. These nanomaterials can also enhance drug efficacy, reduce side effects and improve patient compliance. Despite the promising results, there are still challenges to be addressed in the development of nanomaterial-based delivery systems for RA gene therapy, such as ensuring biocompatibility and stability of the nanomaterials, controlling their pharmacokinetics and biodistribution and addressing potential toxicity concerns. In the future, nanotechnology is expected to play an increasingly important role in RA gene therapy as researchers continue to develop more advanced nanomaterials and improve our understanding of their interactions with biological systems.

## 3. Active Targeting for RA Gene Therapy

In order to increase the accumulation of drugs within the diseased area to minimize the adverse effects, some ligands that can bind to molecules expressed or overexpressed, especially on the inflammatory cells, are decorated on the surface of nanoparticles to achieve active targeting for RA gene delivery. Folic acid (FA), which exhibits a high affinity to folate receptor beta (FR-β), especially expressed on activated macrophages, is commonly used as an active targeting ligand for RA gene therapy [41]. Duan et al. developed a hybrid-nanoparticle system comprised of calcium phosphate/liposome, which was conjugated with folic acid to deliver NF-κB targeted siRNA and methotrexate to diseased sites. When adding nanoparticles with folic acid to macrophages treated with or without LPS, enhanced fluorescence in LPS-activated cells was observed compared to that of non-LPS-activated cells. Compared with non-targeted nanoparticles, the presence of FA improved the cellular uptake of LPS-activated cells. These two observations showed that folic acid could specially bind to FR-β expressed on activated macrophages to reduce the off-target side effects [42]. Methotrexate (MTX) can be used as a therapeutic agent for RA as well as an FR-targeted ligand. Hao et al. designed polymer hybrid micelles (M-PHMs) using two functional amphiphilic polymers (MTX-PEI-LA and mPEG-LA) to encapsulate microRNA-124. The results of cellular uptake demonstrated that M-PHMs increased the cellular uptake of LPS-induced RAW 264.7 cells compared to PHMs without the MTX conjugate. Furthermore, no significant difference was observed in the cellular uptake of M-PHMs compared to PHMs in non-activated macrophages, and the cellular uptake of M-PHMs in activated macrophages could be inhibited by adding free FA, which implied that M-PHMs entered FR-highly expressed macrophages mainly via FR-mediated endocytosis. The in vivo biodistribution also showed that the percentage of fluorescence intensity at the inflammatory tissues in the M-PHMs group was 1.3-fold higher than that of the PHMs, which confirmed that M-PHMs could efficiently deliver the cargo to the inflamed tissues with specificity [43]. Hyaluronic acid (HA) possesses outstanding biocompatibility and high affinity to the CD44 receptor, which is highly expressed in synovial lymphocytes, macrophages and fibroblasts of patients with RA [2]. Li et al. developed HA-coated pH-responsive nanoparticles loaded with MCL siRNA and dexamethasone (HNPs/MD) for RA treatment, and this acid-sensitive delivery system modified with HA could deliver the therapeutic agents for the treatment targeting RA effectively (Figure 4) [44]. There are some other kinds of active targeting ligands used for RA gene therapy, and they are summarized in Table 2. Although active targeting can minimize the risk of off-target effects, there are still some challenges that need to be solved before this strategy can be used widely in RA gene delivery. The scientists need to select ligands with better specificity and lower toxicity, and we should also notice that there may be variability in the markers or signals that are present in different patients or different stages of the disease. More research is necessary to optimize the effectiveness and safety of active targeting, which has the potential to be a valuable strategy in gene therapy for RA.

## 4. siRNA Delivery

siRNAs, which are short 21–23 base pair double-stranded RNA fragments, possess the ability to exert RNA interference (RNAi). The sequence of the siRNA is totally complementary to the mRNA it targets, which directly influences the interpretation of the mRNA and thus reduces the expression of proteins related to the disease pathogenesis [49]. Three stages are involved in the process of RNAi (Figure 5): (1) the siRNA precursor generated in the nucleus travels to the cytoplasm, where it is processed by Dicer enzyme into a double-stranded RNA fragment. (2) A RNA-induced silencing complex (RISC) is formed in the cytoplasm, and the double-strand is unwound by the helicase into a single strand, where the guide strand is combined with the target mRNA via base-pairing. Then, the target mRNA is cleaved and degraded by the RISC endonuclease, which silences the target gene at the transcription level. (3) siRNA and mRNA are combined and act as primers to produce double-strand RNA again, and then the initial stage and effect stage are repeated to further amplify the role of RNAi, and finally, the target mRNA is degraded totally [1]. Since lots of cytokines and other kinds of proteins play an important role in the occurrence and development of RA, RNA interference caused by siRNA may be a potential therapy for RA. Therefore, we will discuss the siRNA delivery systems used in the treatment of RA according to the proteins targeted in this section.

### 4.1. siRNA against TNF-α

Tumor necrosis factor-alpha (TNF-α) plays a vital role in RA inflammation. It is mainly secreted by monocytes and macrophages, but T-cells, B-cells and fibroblasts can produce TNF-α as well [51]. In RA, TNF-α can enhance the proliferation of macrophages, activated T-cells, B-cells and synovial lining cells. It is not only an autocrine stimulator, but also a powerful paracrine inducer of some other inflammatory cytokines, including IL-1, IL-6, IL-8 and granulocyte monocyte-colony stimulating factor (GM-CSF). TNF-α can activate fibroblasts, promote epidermal hyperplasia and recruit inflammatory cells. Following activation by TNF-α and other cytokines, including IL-1 and IL-6, cathepsins and matrix metalloproteinases (MMPs) are overexpressed by synovial fibroblasts. Then, it is followed by collagen and proteoglycan, causing the destruction of cartilage and bone, and eventually, joint erosion happens [51,52]. The concentration of TNF-α in the synovial fluid of RA patients is increased significantly, and the use of TNF-α antagonists such as etanercept, infliximab and adalimumab can relieve and prevent the progression of RA [52]. Although the application of TNF-α antagonists has achieved great success in RA treatment, these therapies can induce some unwanted side effects, such as erythema, nausea and infections [53]. Due to its specificity, potency, flexibility and reversibility, the delivery of siRNA may be a potential strategy for RA treatment, and much research has been carried out on this subject. Aldayel et al. designed an innovative acid-sensitive PEGylated solid-lipid nanoparticle formulation composed of DOTAP, lecithin, cholesterol and acid-sensitive stearic acid-polyethylene glycol (2000) hydrazone conjugate (PHC) to achieve TNF-α siRNA delivery. This delivery system possessed the high encapsulation efficiency of the siRNA (93 ± 2%) and a minimum burst release of the cargo (only approximately 5% of siRNA was released during a month). The pH of the chronic inflammation area was relatively lower, and the distribution and retention of TNF-α siRNA in chronic inflammation sites in CIA mice were greatly enhanced by this formulation (Figure 6) [54]. Jiang et al. reported a kind of nanoparticle-stabilized nanocapsules (NPSCs) that were able to deliver siRNA directly to the cytosol, avoiding endocytosis, for the anti-inflammatory treatment. This system showed that effective suppression of LPS-induced inflammation and TNF-α expression was achieved by the delivery of NPSC/siRNA in vivo, which provided a good method for increasing the efficacy of therapeutic siRNA in the treatment of autoimmune disorders such as RA [55]. To avoid some side effects caused by siRNA, some modifications to the structure of siRNA have been developed. Howard et al. used chitosan nanoparticles to deliver 2′-O-Me modified TNF-α DsiRNA, which could prevent the unwanted innate immune effects of RA treatment. It showed that a lower concentration of type 1 IFN in macrophages taken from mice treated with nanoparticles encapsulating modified DsiRNA was revealed compared to both control siRNA and unmodified DsiRNA formulations, which supported that the incorporation of 2′-O-methyl uridine or guanosine nucleotides into the duplex strand could decrease TLR-7 interaction and associated off-target effects [56]. Apart from directly silencing TNF-α by RNA delivery, there are other strategies to realize the anti-TNF-α therapy, such as increasing the expression of the TNF-α receptor or decreasing the level of TNF-α converting enzyme, and we will talk about them in the following sections.

### 4.2. siRNA against Other Cytokines

In addition to TNF-α, many other pro-inflammatory cytokines play a vital role in the occurrence and development of RA. IL-1β possesses the capacity to stimulate immune cells and regulate the expression of pro-inflammatory cytokines. It can promote inflammatory progression and joint destruction by promoting the production of pro-inflammatory mediators such as nitric oxide synthase and cyclooxygenase-2. It can also stimulate synoviocytes to secrete matrix metalloproteinases, which accelerates cartilage damage. Furthermore, it participates in the erosion of the subchondral bone of RA joints via producing receptor activator of the nuclear factor-κB ligand (RANKL) and stimulating the maturation of osteoclasts [1,57]. Mainly secreted by activated macrophages and fibroblast-like synoviocytes, IL-6 plays a role in different inflammatory effector pathways via the activation of immune cells, endothelial cells, synoviocytes or osteoclasts and the production of acute-phase proteins such as CRP [1,58]. IL-18 is also a pro-inflammatory cytokine secreted by antigen-presenting cells and T cells. It plays an important part in the occurrence and maintenance of the inflammatory response during RA, and the expression of IL-18 in synovial tissues correlates with the severity of joint inflammation and the levels of TNF-α and IL-1β [47]. In order to develop an alternative targeted therapy for RA patients who do not respond to the anti-TNF biotherapies, Khoury et al. used RPR209120 (2-[3-(bis-(3-aminopropyl)-amino)-propylamino]-N-ditetradecylcarbamoylmethylacetamide) and DOPE (1,2-dioleoyl-sn-glycero-3-phosphoethanolamine) to form lipoplexes to encapsulate anti-IL-1, anti-IL-6 or anti-IL-18 siRNA. The results showed that beneficial effects were achieved in CIA models by delivering anti-IL-1β siRNA, anti-IL-6 siRNA or anti-IL-18 siRNA nanoparticles alone, and better therapeutic efficacy was observed when using these three kinds of nanoparticles in combination. Meanwhile, this combination was found to be as powerful as TNF siRNA for the treatment of RA, which provided an alternative therapy for patients with refractory RA [45]. Song et al. reported a lipidoid–polymer hybrid nanoparticle composed of Pluronic F127 and spermidine-based lipidoid (S14) to deliver anti-IL-1β siRNA and studied its therapeutic effects in CAIA mice. This system silenced the expression of IL-1β effectively, and the inflammatory symptoms, such as paw swelling and the production of inflammatory cytokines as well as progressive bone destruction, were suppressed. In addition, this treatment could reduce the production of pro-inflammatory cytokines in circulation without influencing the gene expression in non-targeted tissues, which did not disturb the normal physiological function of other tissues [21]. Since IL-15 was involved in the pathogenesis of RA, and immunotoxins targeting IL-15R-bearing cells such as activated macrophages and T cells were reported to relieve disease severity in AA rats [59], Zhang et al. demonstrated polyethylenimine (PEI)-complexed siRNA nanoparticles to target the β chain of IL-15R, which was shared by the receptor for IL-2 (IL-2/15 receptor β chain). The average hydrodynamic diameter was 246 nm, which allowed significant amounts of the nanoparticles to accumulate in the inflammatory joints due to the EPR effect and be taken up by macrophages and T cells efficiently. The symptoms of RA were significantly reduced after the intravenous administration once per week for three weeks, and the expression of TNF-α, IL-1β and MMP-3 was reduced sharply, which confirmed that IL-2/15 receptor β chain could be a potential target for RA treatment (Figure 7) [60].

### 4.3. siRNA Blocking Pathways

Pathways involved in the occurrence and progression of RA may be potential targets for RA gene treatment. However, it should be noted that the occurrence of RA results from the action of multiple signal pathways, and targeting only one pathway usually can not inhibit RA progression. Investigating each pathway thoroughly helps us connect them to design effective blocking drugs. Pathways often involved in RA include NF-κB signaling pathway, JAK/STAT signaling pathway, MAPK signaling pathway, Toll-like receptors signaling pathway and Wnt signaling pathway [1]. The activation of the NF-κB signaling pathway can result in the secretion of various inflammatory cytokines such as TNF-α and IL-1β, which would, in turn, activate NF-κB and further amplify RA inflammatory response [1]. In addition, NF-κB has been found to be involved in the polarization of macrophages, and the level of M2-type macrophages could be enhanced by inhibiting this pathway [18,61]. Therefore, blocking the NF-κB signaling pathway may be a useful way to alleviate the symptoms of RA. Wang et al. designed a polymeric hybrid micelle system composed of PCL-PEI and PCL-PEG to deliver dexamethasone and siRNA targeting NF-κB p65, which decreased the inflammation and changed the ratio of M1 macrophages to M2 macrophages. The particle size was 98.41 ± 1.01 nm, and the zeta potential of the micelle was 4.22 ± 1.34 mV. After treating with this micelle system in vitro, levels of p65 mRNA, TNF-α mRNA and IL-1β mRNA were much lower, indicating its outstanding anti-inflammatory effects in vitro. The treatment with this system reduced the levels of iNOS mRNA and IL-12 mRNA, which were M1 markers and enhanced the levels of mRNAs of M2 markers Arg-1 and CD206, indicating this delivery strategy could repolarize macrophages from the M1 to M2 type. In the experiment investigating the in vivo therapeutic efficacy, the micelles delayed the RA progression and decreased the serum levels of TNF-α and IL-1β. Furthermore, the mildest histopathology was observed in the group treated with micelles co-loaded with dexamethasone and siRNA without affecting renal or liver function [18]. Duan et al. reported a hybrid-nanoparticle system composed of calcium phosphate/liposome to deliver p65 siRNA and methotrexate and decorated the nanoparticle surface with folic acid, which targets the FRβ in the activated macrophages [42]. Kanazawa et al. delivered p65 siRNA using a novel micelle composed of MPEG-PCL-CH2R4H2C. This oligopeptide-modified polymer, MPEG-PCL-CH2R4H2C, which was built of arginine, histidine and cysteine, could complex with siRNA and deliver it to the inflammatory sites. This complex showed better effects in CIA mice compared with MTX and could prevent synovium erosion following the injury of cartilage, implying a potential strategy for RA treatment (Figure 8) [62]. Zhou et al. reported nanoparticles composed of the melittin-derived p5RHH peptides to deliver anti-p65 siRNA, which exhibited the potential ability to suppress ongoing experimental arthritis. About 55 nm nanoparticles could be formed within 10 min via noncovalent self-assembly and could be injected directly, which effectively repressed early inflammatory arthritis without influencing the expression of p65 in off-target tissues or inducing a humoral response after serial administration [38]. HIF-1α transcription factor plays an important role in the regulation of the cellular response to hypoxia. It can be activated in response to hypoxia as well as an inflammatory microenvironment to increase the expression of various genes modulating metabolism, angiogenesis and inflammatory response. In addition, NF-κB and MAPK members take part in HIF-1α activation and stabilization under hypoxic conditions or stimulated with bacterial products. Liu et al. developed HIF-1α siRNA-loaded calcium phosphate nanoparticles encapsulated in apolipoprotein E3-reconstituted high-density lipoprotein (HIF-CaP-rHDL) for RA treatment. As they were similar to the inorganic materials in natural bone, CaP-based biomaterials were often used for bone repair, and the calcium ions released from these materials could attenuate the inflammatory response of macrophages by repressing the activated inflammatory NF-κB and MAPK signaling pathways. Additionally, apoE3 used in this system could exhibit anti-inflammatory, antioxidant and antithrombotic effects. The nanoparticles showed excellent anti-inflammatory effects following HIF-1α knockdown and NF-κB and MAPK signaling pathway inhibition in LPS-activated macrophages. They could also inhibit the receptor of NF-κB ligand (RANKL)-induced osteoclast formation and alleviate bone erosion, cartilage damage and osteoclastogenesis [63]. Apart from blocking NF-κB, Scheinman et al. fabricated an RGD peptide-functionalized PLGA nanosystem to deliver a STAT1 siRNA to joint tissues. Since the transcription factor STAT1 was the primary mediator of IFN-γ signal transduction, the researchers believed that the central amplification network within the arthritic joint could be down-regulated by inhibiting the expression of STAT1 [48]. Feng et al. screened siRNAs targeting the endoplasmic reticulum to the nucleus signaling 1 (ERN1) gene and used PEI and poly (β-amino amine) (PBAA) to deliver them to the macrophages. They synthesized the disulfide-containing cationic core ss-PBAA-PEI (NPs) and coupled a cell-penetrating peptide RKKRRQRRR (R) to the core and then formed R-NPs. After siRNAs were loaded onto the R-NPs, folic acid (FA) was modified to the surface of the nanoparticles through the PEG shell. The hydrodynamic size of the nanoparticle was 203.3 nm, and it was verified that the nanoparticle possessed good biocompatibility and pH/redox responsiveness. The siERN1 could modulate the levels of the intracellular calcium ions by interfering with the function of inositol 1,4,5-trisphosphate receptor 1/3 (IP3R1/3) and thus resulting in M2 polarization of macrophages. This formulation also acted as a conductor of macrophage polarization by controlling the levels of the calcium ions and was an inhibitor of MyD88-dependent Toll-like receptor signaling [64].

In addition to the therapeutic mentioned above, some other potential strategies were also discovered in recent years, such as reducing the expression of matrix metalloproteinases [17], anti-COX-2 therapy [65], inhibiting the transcription of Bruton’s tyrosine kinase (BTK) genes [66] and so on. The details of such research are listed in Table 3.

## 5. DNA Delivery

Delivering DNA using a nanoparticle delivery system to express certain proteins may be a useful method for RA treatment. In order to realize efficient delivery, DNA delivery systems must achieve effective nuclear entry in addition to the challenges that the siRNA delivery system needs to conquer, which can be solved by adding nuclear localization signal peptides [75]. The anti-inflammatory cytokine IL-10 is found to be a potential target for RA DNA delivery. Secreted mainly by macrophages in the rheumatoid synovium, IL-10 can inhibit the production of inflammatory cytokines such as TNF-α, IL-1β and IL-6 [76]. Exogenous IL-10 can also protect articular cartilage and inhibit the migration of monocytes into the joint synovium [1]. Furthermore, macrophage polarization can be skewed to an M2 phenotype as well [5], which may provide a novel strategy for RA treatment. However, due to its poor stability and short half time in vivo, the application of IL-10 delivery in clinical trials is limited, which can be solved by IL-10 plasmid delivery. Jain et al. developed tuftsin-modified alginate nanoparticles encapsulating mIL-10 plasmid DNA for the treatment of experimental arthritis. Alginate-formed cross-linked nanoparticles in the presence of Ca^2+^ and tuftsin peptides were modified on the surface, which could increase the cellular uptake of macrophages. This delivery could effectively enhance the expression of mIL-10 and down-regulate the expression of TNF-α [77]. The in vivo experiment showed that these nanoparticles could reprogram the balance of macrophage phenotype, decrease the expression of systemic and joint tissue pro-inflammatory cytokines and protect the joint from damage, offering an innovative idea for the treatment of RA and other chronic inflammatory diseases [72]. Zheng et al. reported a biomimetic delivery system using human serum albumin (HAS) to co-deliver IL-10 pDNA and dexamethasone sodium phosphate (DSP). In addition to advantages, such as water solubility, non-immunogenicity and biodegradation, HAS also showed a high affinity for the glycoprotein SPARC, which was overexpressed in the inflammatory microenvironment in RA. The results revealed that the nanoparticles could be accumulated in the inflammatory sites and could relieve joint swelling and bone erosion by reducing the production of pro-inflammatory cytokines and increasing the expression of IL-10, which was able to promote M1-to-M2 macrophage repolarization [40]. Apart from HAS, M2-type exosomes could be used to co-deliver IL-10 pDNA and betamethasone sodium phosphate (BSP) for RA treatment as well, which could not only regulate the balance of M1/M2 macrophages, but could also decrease the side effects of BSP [27]. Aside from silencing TNF-α directly by RNAi, increasing the levels of tumor necrosis factor (TNF)-α receptor (TNFR) may also be a good idea for anti-TNF-α treatment. Wang et al. used ultrasound-targeted microbubble destruction (UTMD) for the transfection of TNFR plasmid, and the TNFR gene was observed to be continuously expressed in the rats for 8 weeks at least, which might alleviate RA symptoms and decrease the concentrations in the synovial tissues and peripheral blood [73].

Some non-coding nucleotides are used for RA gene therapy as well. For example, Hao et al. designed a co-delivery system composed of MTX-PEI-LA and mPEG-LA for the combination of MTX and microRNA-124, which possessed direct bone activity against RA [43]. However, the research on this subject is still limited, and we summarized the existing work in Table 3.

## 6. Strategies Targeting cfDNA

It was found that the levels of circulating cell-free DNA (cfDNA) in RA patients were much higher than that in normal people, and it was exclusive to RA patients that the concentrations of cell-free nuclear and mitochondrial DNA in the plasma were many folds lower than corresponding synovial fluid levels [46]. Toll-like receptors on macrophages and neutrophils can be activated by cfDNA and then secrete inflammatory factors such as TNF-α and IL-6. In addition, cfDNA antibodies can form immune complexes with cfDNA, further promoting B cells to proliferate and produce inflammatory cytokines. Therefore, reducing the levels of cfDNA directly from the RA microenvironment can keep it away from binding to different immune cells, such as macrophages, B cells and neutrophils, which may be effective for RA treatment [78]. As cfDNA is highly negatively charged, nucleic-acid-binding nanomaterials with rich positive charges can be applied to remove cfDNA [79,80]. Liang et al. designed ~40 nm cationic nanoparticles with high binding affinity for cfDNA by using the diblock copolymer of poly (lactic-co-glycolic acid) (PLGA) and poly(2-(diethy-amino)ethyl methacrylate) (PDMA), which could inhibit the activation of primary synovial fluid monocytes and fibroblast-like synoviocytes and alleviate the symptoms of RA such as tissue swelling, bone erosion and cartilage damage [81]. In order to decrease the toxicity of the cationic nanomaterials and elevate the therapeutic efficacy, Liu et al. modified the cationic nanoparticles with MMP2-sensitive peptide-linked PEG (cNP-pp-PEG). Since MMP2 was enriched in the inflamed joints, PEG could only be removed in the inflammatory sites and then the cations were exposed, which reduced the systemic toxicity during circulation after intravenous administration. They also loaded methotrexate (MTX) into the core of the cNP-pp-PEG via hydrophobicity interaction, which strengthened the anti-inflammatory efficacy of cNPs [82]. Taking advantage of polydopamine’s (P) ideal characteristics, including favorable biocompatibility, biodegradability and good bioadhesion, Chen et al. decorated P with dimethylamino groups to form altered charged DPs to increase the biocompatibility and scavenging efficacy of the scavenger. They also found that the DPs with a higher degree of substitution of bis dimethylamino group showed a higher density of positive charge and stronger cfDNA binding affinity, resulting in better RA therapeutic efficacy [83]. With the development of nanomaterials and intelligent drug delivery systems, cfDNA scavengers might act as useful tools for more effective RA treatment.

## 7. Challenges and Perspectives on the Nucleic Acid Delivery System for Rheumatoid Arthritis Therapy

Although there has been lots of research on nucleic acid delivery systems for rheumatoid arthritis therapy, there are some limitations to be noted:Limited tissue specificity: One of the challenges of nucleic acid delivery is achieving specific targeting to the inflammatory tissues in rheumatoid arthritis, which can limit the therapeutic effects of the nucleic acids. Furthermore, the complexity of the joint environment and the rapid clearance from the body may increase the difficulty of precise targeting.Limited cell entry: One of the challenges of nucleic acid delivery is getting the genetic material into the cells where it can have an effect. Many cells have mechanisms to protect against foreign nucleic acids, which can make it difficult for the therapeutic nucleic acids to enter the cells. The efficiency of cell entry can be improved by constructing biomimetic nanoparticles or modifying the surface of the delivery systems, such as changing the surface charges or adding active targeting ligands.Short-lived effects: As the nucleic acids used for gene therapy do not integrate into the cell genome, their therapeutic efficacy will weaken along with the division and death of cells. Therefore, the effects of gene therapy may be short-lived, particularly if the cells receiving the nucleic acid are rapidly dividing. This can limit the effectiveness of the therapy and require multiple treatments over time. Since RA is a chronic disease that needs long-term treatment, novel delivery systems, such as hydrogel systems, with sustained release capabilities can prolong the efficacy of therapeutic genes [84].Immune response: The body’s immune system may react to the therapeutic nucleic acids as foreign substances, potentially leading to an immune response that reduces the effectiveness of the therapy or causes side effects.Off-target effects: Although active targeting strategies are applied to nucleic acid delivery systems, they may still have off-target effects, leading to unintended changes in gene expression or other cellular processes. This could potentially cause harm or limit the effectiveness of the therapy.Patient individuality: As the etiology of RA is complex and the cause in each patient is different, the response to the same gene therapy drug varies among each patient. More clinical research is needed to optimize the therapeutic approach and determine which patients may benefit most from gene therapy.Long-term effects: The long-term effects of gene therapy for rheumatoid arthritis are not yet fully understood. As with any new therapy, there is a need for ongoing monitoring of safety and efficacy over time to determine the potential benefits and risks of treatment.Cost and accessibility: Gene therapy can be expensive and may not be accessible to all patients due to cost or availability issues.

Considering the limitations mentioned above, the development of the delivery systems used for RA gene therapy should focus on the following:Improved targeting: The development of more targeted delivery systems could improve the specificity and efficiency of gene delivery to cells and tissues involved in rheumatoid arthritis.Combination therapy: Nucleic acid delivery systems may be used in combination with other rheumatoid arthritis therapies, such as DMARDs or biologics, to enhance efficacy and reduce side effects. Combining nucleic acid delivery systems with immunotherapies could enhance the immune response to the therapeutic gene and improve the overall effectiveness of rheumatoid arthritis treatment.Localized delivery: The use of localized delivery systems, such as injectable hydrogels, could enable targeted delivery of therapeutic nucleic acids to specific joint sites affected by rheumatoid arthritis.Personalized medicine: Gene therapy has the potential to be tailored to each individual patient based on their specific genetic and disease characteristics. This could lead to more effective and targeted treatments with fewer side effects.Safety: The long-term effects of gene therapy for rheumatoid arthritis are not yet fully understood. As with any new therapy, there is a need for ongoing monitoring of safety and efficacy over time to determine the potential benefits and risks of treatment.

It is important to note that the development and optimization of nucleic acid delivery systems for RA gene therapy is an ongoing area of research. Further studies, clinical trials and close collaboration between patients, researchers and healthcare providers will be necessary to fully understand the potential benefits and limitations of these systems for rheumatoid arthritis treatment.

## 8. Conclusions

RA treatment is still full of challenges, and it has attracted many researchers’ interest in looking for new therapies. Although various drugs have been developed, there are still some challenges, such as serious side effects, that need to be solved. With the development of gene delivery, RA gene therapy may be an alternative treatment for patients who do not show ideal responses to the existing drugs. In this review, we have summarized the innovative nanomaterials and active targeting ligands used for RA gene therapy and listed the potential therapeutic genes and the corresponding gene delivery system being studied in recent years.

Due to the different advantages and disadvantages of each gene delivery system, it is difficult to determine which delivery system is most suitable for RA gene therapy. However, in future gene therapy for RA, research should focus on the following points. First, in order to select the best therapeutic gene, we need to fully understand the pathogenesis of RA and explore the differences in therapeutic effects of the gene in different patients through extensive preclinical and clinical studies to choose the appropriate therapeutic gene according to the actual situation of different patients, thus achieving precision medicine. Secondly, because the pathogenesis of RA is complex and it is induced by the interaction of different pathways and cytokines, it is hard to delay the RA progression by targeting a single gene alone. Therefore, combination therapy should be adopted, such as gene combination therapy, targeting different targets or the co-delivery of genes and chemical drugs together, which might accomplish better therapeutic efficacy and fewer side effects. In terms of gene delivery system construction, we should select appropriate nanomaterials and delivery strategies based on the characteristics of the target delivery substance and the targeting sites in vivo to achieve more precise and efficient gene delivery. An ideal nucleic acid delivery system should have the following characteristics: good protection of nucleic acid substances, good stability, good targeting specificity, good cellular uptake efficiency, effective lysosomal escape, high transfection efficiency and good biosafety. In addition to the selection of the target gene and the design of the delivery systems, we should also pay attention to the administration mode, dosage and frequency of RA gene therapy. As a systemic disease, systemic administration is more ideal than local administration, but it may not be as specific as local administration, such as intra-articular injection and systemic administration may cause off-target effects and unwanted side effects. For rheumatoid arthritis, a chronic disease that requires long-term treatment, the dosage and frequency of administration should be carefully considered based on a large number of preclinical and clinical studies. To reduce the frequency of administration and improve patient compliance, some gene delivery systems with controlled release may be more suitable.

In conclusion, RA gene therapy is a promising tool for the treatment of RA in the near future. However, significant efforts are still needed to overcome the challenges associated with gene delivery and ensure the safe and effective use of this therapy. Ongoing research in this area is likely to lead to the development of more effective treatments for RA gene therapy in the years to come.

## Figures and Tables

**Figure 1 pharmaceutics-15-01237-f001:**
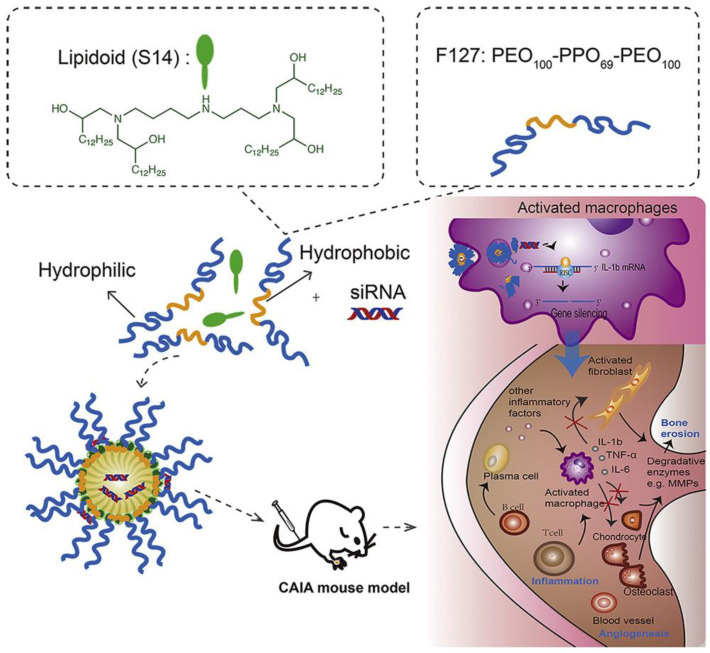
Schematic preparation and mechanism of lipidoid-polymer hybrid nanoparticles with anti-inflammatory effects. Reprinted with permission from Ref. [21]. Copyright © 2019 The American Society of Gene and Cell Therapy.

**Figure 2 pharmaceutics-15-01237-f002:**
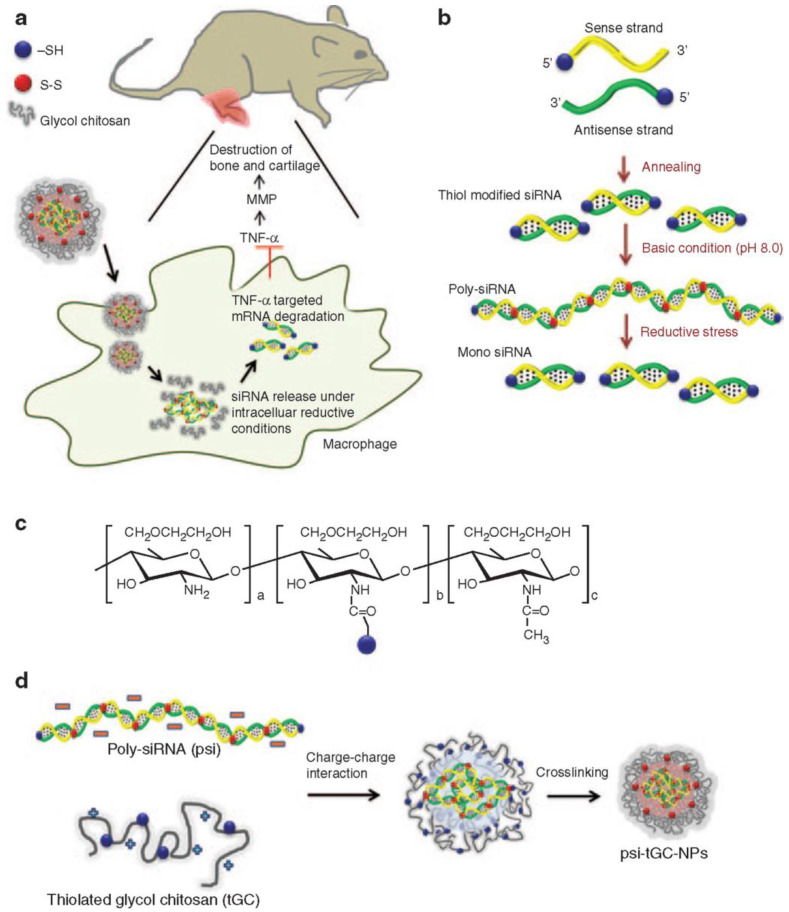
Polymerized siRNA/thiolated glycol chitosan nanoparticles (psi-tGC-NPs). (**a**) Cellular uptake of psi-tGC-NPs into macrophages resulting in TNF-α gene salience. (**b**) Preparation of poly-siRNA. (**c**) Structure of tGC polymers. (**d**) Complexation of poly-siRNA with tGC polymers. Reprinted with permission from Ref. [23]. Copyright © 2014 The American Society of Gene and Cell Therapy. Published by Elsevier Inc. All rights reserved.

**Figure 3 pharmaceutics-15-01237-f003:**
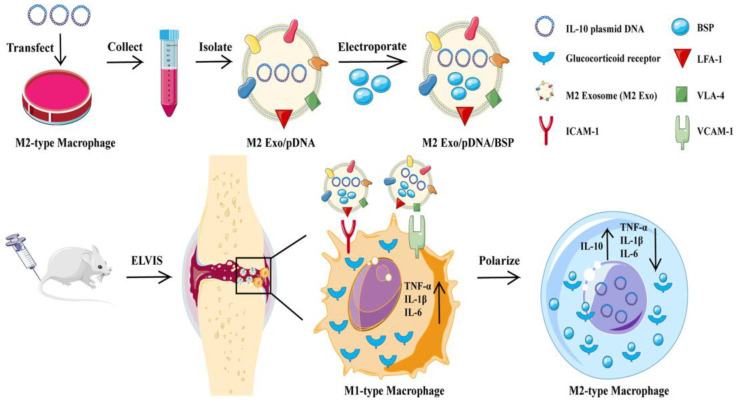
Schematic preparation, cellular uptake and mechanism of Exo/pDNA/BSP with the ability to regulate the polarization of macrophages. Reprinted with permission from Ref. [27]. Copyright © 2021 Published by Elsevier B.V.

**Figure 4 pharmaceutics-15-01237-f004:**
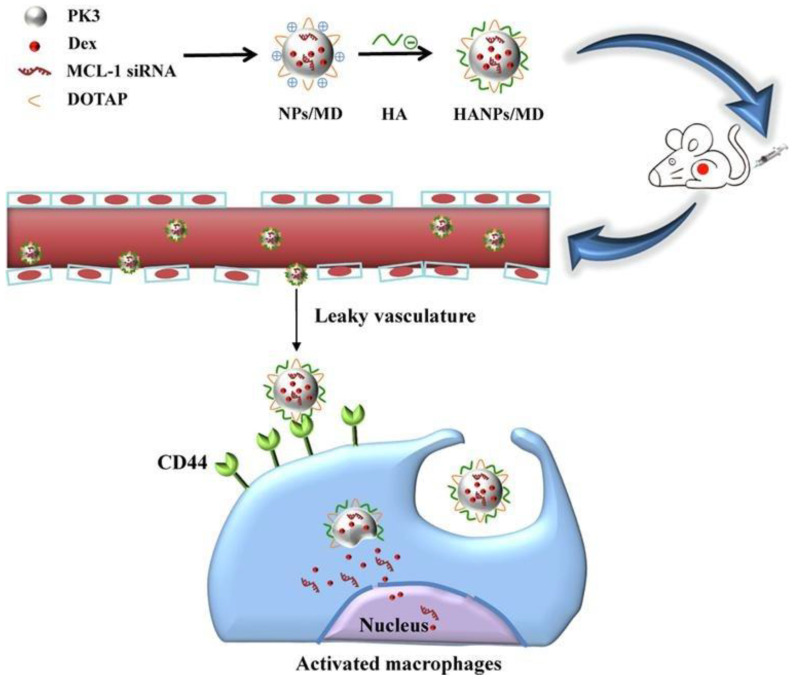
Schematic preparation and cellular uptake of HANPs/MD with high affinity to macrophages. Reprinted with permission from Ref. [44]. Copyright © 2020 Elsevier B.V. All rights reserved.

**Figure 5 pharmaceutics-15-01237-f005:**
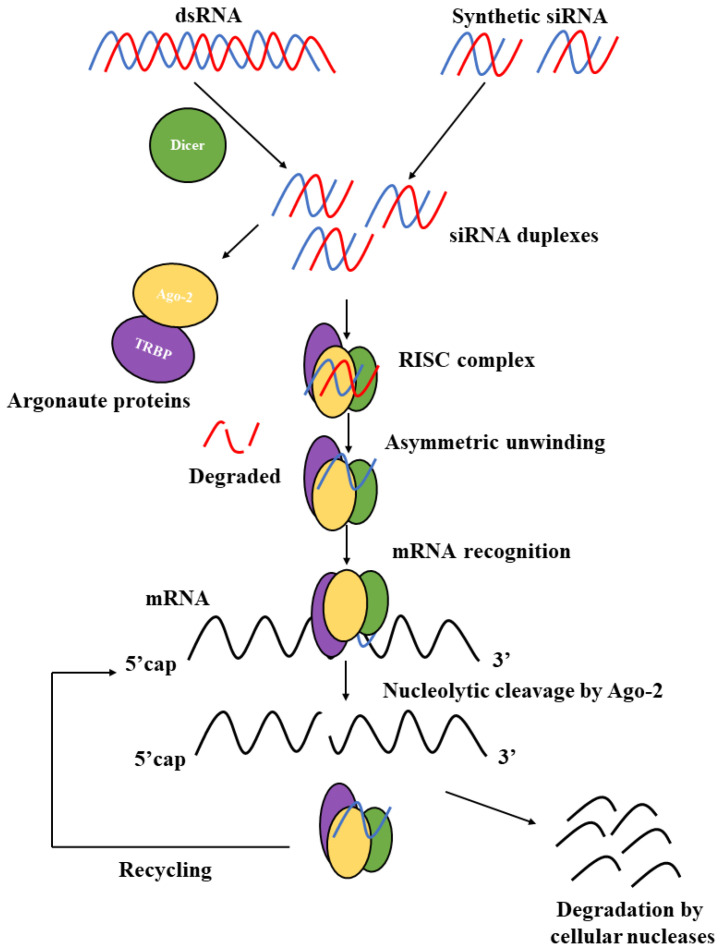
Schematic illustration of the mechanism of gene silencing by double-stranded RNAs [50].

**Figure 6 pharmaceutics-15-01237-f006:**
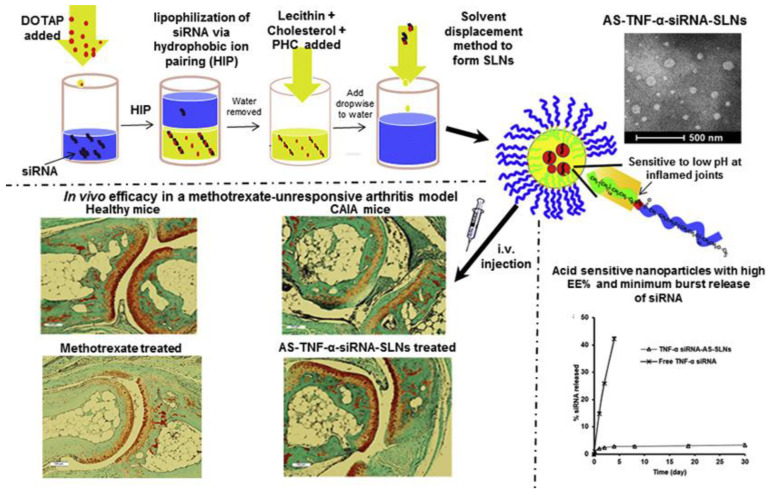
Schematic preparation, characterization and in vivo efficacy of AS-TNF-α-siRNA-SLNs. Reprinted with permission from Ref. [54]. Copyright © 2018 Elsevier B.V. All rights reserved.

**Figure 7 pharmaceutics-15-01237-f007:**
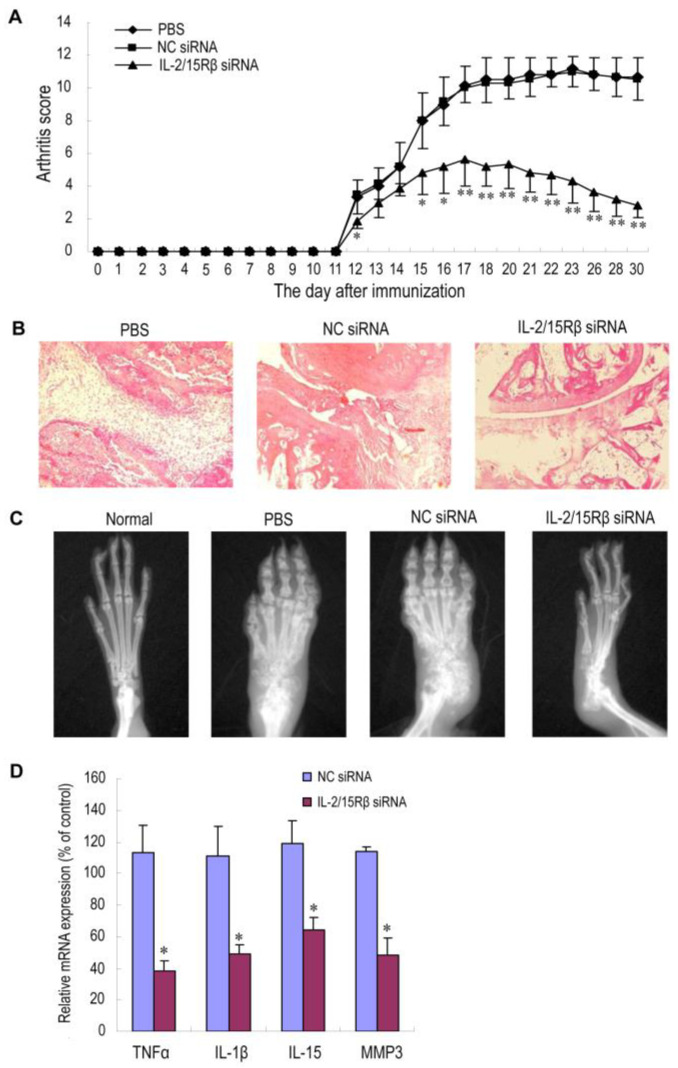
Therapeutic effects of PEI/IL-2/15Rβ siRNA complexes on AA rats. Male Wistar rats (n = 6 per group) were given Freund’s complete adjuvant in the left hind footpad on day 0. IL-2/15Rβ siRNA-5 polyplexes were intravenously administered on days 10, 17 and 24 at 0.3 mg siRNA/kg. The rats receiving NC siRNA polyplexes or PBS served as controls. (**A**) Arthritis score. Values are the mean and SD. * *p* < 0.05; ** *p* < 0.01, versus the PBS group. (**B**) Representative histopathologies of the right hind ankle joints stained with hematoxylin and eosin (H&E). Rats were sacrificed and subjected to histopathological examination on day 31. Minimal articular inflammation and joint destruction were observed in the group treated with IL-2/15Rβ siRNA polyplexes, whereas PBS- and NC siRNA-treated rats exhibited pronounced synovial hyperplasia, bone damage and joint space narrowing. (**C**) Representative radiographs of right hind paws (n = 3 per group). On day 31, the rats were anesthetized and subjected to radiography. Neither paw swelling nor joint damage was observed in normal rats. Severe bone erosion was seen in AA rats treated with either PBS or NC siRNA polyplexes, but the damage was much less in rats treated with IL-2/15Rβ siRNA polyplexes. (**D**) Effects of IL-2/15Rβ siRNA polyplexes on mRNA expression of inflammatory factors. At the end of the experiment (day 31), RNA was prepared from the right hind ankle joints of AA rats in each group. mRNA levels of each factor were determined by qPCR, normalized against GAPDH, and expressed as percentage of the PBS control. The result is presented as values (the mean and SD) obtained from three different samples randomly selected in each group. * *p* < 0.05, versus PBS control. Reprinted with permission from Ref. [60]. Copyright © 2013 Zhang et al.

**Figure 8 pharmaceutics-15-01237-f008:**
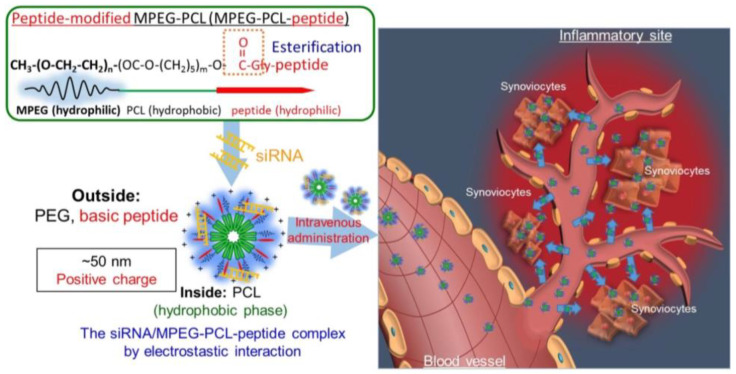
Schematic preparation, characterization of the complexes with MPEG-PCL-CH_2_R_4_H_2_C and their biodistribution in vivo. Reprinted with permission from Ref. [62]. Copyright © 2016 Elsevier B.V. All rights reserved.

**Table 1 pharmaceutics-15-01237-t001:** Nanomaterials for RA gene therapy.

Nanomaterials	Advantages	Limitations	Delivery System	Refs.
Polyethylenimine	High transfection efficiency; pH buffering capacity; Endosomal escape capacity; Ability to transfect non-dividing cells	Potential toxicity; Inflammatory response	A in situ hydrogel loaded with disulfide-crosslinked polyethyleneimine nanoparticles	[17]
			A hybrid micelle system consisting of PCL-PEI and PCL-PEG	[18]
Lipids	Low immunogenicity; Good biocompatibility; Versatility	Low stability; Short half-time in the bloodstream; Potential toxicity	Liposomes consisting of RPR209120 and DOPE	[19]
			Lipid-polymer hybrid nanoparticles consisting of DOTAP and PLGA	[20]
			Lipidoid-polymer hybrid nanoparticles consisting of Pluronic F127 and spermidine-based lipidoid (S14)	[21]
Chitosan	Good biocompatibility; Low toxicity; Cationic charge Ability to be modified with active targeting ligands	Poor water solubility; Poor targeting ability; Charge deduction under physiological condition; Premature release in the cytoplasm	Nanoparticles consisting of a chitosan derivative containing diethylethylamine (DEAE)	[22]
			A nanocomplex consisting of thiolated glycol chitosan	[23]
Iron oxide nanoparticles	Biocompatibility; Magnetic property; High surface area; Biodegradability; Imaging capability	Limited loading capacity; Instability; Limited specificity	Polyethyleneimine-functionalized iron oxide nanoparticles	[23]
Cationic peptides	High delivery efficiency; Low immunogenicity; Specificity; Good biocompatibility; Ease of synthesis	Potential toxicity; Instability; Limited loading capacity	Anti-TNF-α siRNA complexed to RVG-9R	[24,25,26]
Biomimetic nanoparticles	Enhanced targeting; Good biocompatibility; Versatility; Reduced immunogenicity	Complexity; Limited loading capacity; High cost	M2 exosomes	[27]

**Table 2 pharmaceutics-15-01237-t002:** Active targeting ligands for RA gene treatment.

Type	Receptor	Refs.
Folic acid	Folate receptor β	[32,38,45,46]
Methotrexate	Folate receptor β	[39]
RGD	αvβ3 integrins	[47]
Tuftsin peptide	Fc and neuropilin-1 receptors	[48]
D-Asp8	Bone resorption surfaces	[35]
Human serum albumin	Synovial tissues	[37]
Hyaluronic acid	CD44	[25]

**Table 3 pharmaceutics-15-01237-t003:** Gene delivery systems for RA treatment.

Type	Nucleic Acids	Delivery System	Co-Delivery	Administration	Ref.
siRNA	MMP9 siRNA	In situ hydrogel loaded with PEI-SS-IND-MTX-MMP-9 siRNA NPs	Indomethacin and methotrexate	i.a.	[17]
siRNA	IL-2/15Rβ siRNA	PEI/siRNA complexes	-	i.v.	[60]
siRNA	IL-2/15Rβ siRNA	Polyethyleneimine -superparamagnetic iron oxide nanoparticle	-	i.v.	[37]
siRNA	anti-COX2 siRNA	Drug-loaded PLGA nanoparticles complexed with poly (-ethyleneimine) (PEI)/siRNA	Dexamethasone	-	[65]
siRNA	NF-kB-targeted siRNA (p65 siRNA)	Calcium phosphate/liposome	Methotrexate	i.v.	[42]
siRNA	NF-kB-targeted siRNA (p65 siRNA)	Polymeric hybrid micelle (98% PCL-PEG and 2% PCL-PEI)	Dexamethasone	i.v.	[18]
siRNA	NF-kB-targeted siRNA (p65 siRNA)	Arginine–histidine–cysteine-based oligopeptide-modified polymer micelles	-	i.v.	[62]
siRNA	NF-kB-targeted siRNA (p65 siRNA)	Melittin-derived cationic amphipathic peptide-siRNA nanocomplexes	-	i.v.	[38]
siRNA	siRNA targeting Bruton’s tyrosine kinase	Cationic lipid-assisted PEG-b-PLGA nanoparticles	-	i.v.	[66]
siRNA	Several anti-cytokine siRNA (anti-IL-1, anti-IL-6, anti-IL-18)	Lipoplexes	-	i.v.	[45]
siRNA	STAT1 siRNA	RGD-PLGA nanoparticles	-	i.v.	[48]
siRNA	Notch1 targeting siRNA	Thiolated glycol chitosan (tGC) nanoparticles	-	i.v.	[67]
siRNA	IL-1β siRNA	Lipidoid-polymer hybrid nanoparticle	-	i.v.	[21]
siRNA	HIF-1α siRNA	HIF-1α siRNA-loaded calcium phosphate nanoparticles encapsulated in apolipoprotein E3-reconstituted high-density lipoprotein	-	i.v.	[63]
siRNA	Ribonucleotide reductase M2 (RRM2) siRNA	Cell permeable peptide-conjugated liposome/protamine/DNA/RRM2 siRNA complex	-	-	[68]
siRNA	siERN1	FA(folic acid)−PEG−R(RKKRRQRRR)−NPs(ss−PBAA−PEI)@siERN1	-	i.v.	[64]
siRNA	Myeloid cell leukemia-1 (Mcl-1) siRNA	Polymeric nanoparticles composed of PK3 as a pH-sensitive polymer, folate-polyethyleneglycol-poly(lactide-co-glycolide) as a targeting ligand and a DOTAP/siRNA core	-	i.v.	[69]
siRNA	Mcl-1 (siRNA)	HA-coated and pH-responsive nanoparticles	Dexamethasone	i.v.	[44]
siRNA	TNF-α siRNA	Nanoparticle-stabilized nanocapsules	-	i.v.	[55]
siRNA	TNF-α siRNA	Acid-sensitive sheddable PEGylated solid-lipid nanoparticle	-	i.v.	[54]
siRNA	TNF-α siRNA	Lipidpolymer hybrid nanoparticles	-	i.a.	[20]
siRNA	TNF-α siRNA	RVG-9R/siRNA complexes	-	i.v.	[24]
siRNA	TNF-α siRNA	PLGA nanoparticles (DOTAP-modified)	-	i.a.	[70]
siRNA	TNF-α siRNA	Nanocomplexes of polymerized siRNA (poly-siRNA) targeting TNF-α with thiolated glycol chitosan (tGC) polymers	-	i.v.	[23]
siRNA	TNF-α siRNA	siRNA/WS complex		i.v.	[71]
siRNA	TNF-α siRNA	Folate-PEG-chitosan-DEAE/siRNA nanoparticles	-	i.p.	[22]
siRNA	TNF siRNA	Lipid-polymer hybrid nanoparticles (LPNs) composed of lipidoid and poly (DLlactic-co-glycolic acid)	-	i.a.	[20]
siRNA	anti-TNF-α Dicer-substrate siRNA (2′-OMe-modified)	Chitosan/siRNA nanoparticles	-	i.p.	[56]
DNA	IL-10 plasmid DNA	M2-type exosomes nanoparticles	Betamethasone sodium phosphate	i.v.	[27]
DNA	IL-10 plasmid DNA	Tuftsin-modified alginate nanoparticles	-	i.p.	[72]
DNA	IL-10 plasmid DNA	Human serum albumin (HSA) preparing pDNA/DSP-NPs	Dexamethasone sodium phosphate	i.v.	[40]
DNA	TNF-α receptor (TNFR) gene	Ultrasound-targeted microbubble	-	Injected at the ankle joint and tibialis anterior muscle, respectively	[73]
Non-coding nucleotide	miRNA-124	Methotrexate-conjugated polymer hybrid micelles	Methotrexate	i.v.	[43]
Non-coding nucleotide	TNF-α short hairpin RNA (shRNA) (Plasmids pIN27-hU6- TNF-α-shRNA)	Oral yeast microcapsules	-	Oral administration	[74]
Non-coding nucleotide	TNF-α converting enzyme (TACE) (shRNA) in the psi-U6.1 vector driven by the U6 promoter	shTACE/peptide (8D-16R) carrier complex	-	i.v.	[39]

## Data Availability

No new data were created or analyzed in this study. Data sharing is not applicable to this article.

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
