# Peer review of "Research Advances in Nucleic Acid Delivery System for Rheumatoid Arthritis Therapy"

_pharmaceutics, 2023, doi:10.3390/pharmaceutics15041237_

Round 1
Reviewer 1 Report
The review summarized the existing nanomaterials and active targeting ligands used for RA gene therapy and also introduced various gene delivery systems for RA treatment which may enlighten the relative research in the future. However, some of the concerns should be addressed before getting accepted for publication.
-> Figure 1, 2, 3, and 6 as well as the text in those figures seems less clear and legible, please revise them keeping more clear and readable figures.
-> Please explain in the background/introduction why this review is different from other recently reported similar reviews.
->The subsections should be there for the drawbacks/limitations and future outlook.
-> What are authors' thoughts, among the different therapeutics/nucleic acid delivery systems discussed, which one is most suitable or improved for the rheumatoid arthritis therapy?
->The introduction and discussion section also demand some more input and corrections. Especially, the discussion section has missed many sections regarding inputs from the authors as well as referring to previous literature.
->The grammatical errors and typing errors should be rechecked, therefore revise the manuscript to eradicate all those mistakes.
-> Some in-depth concluding remarks should be added by keeping the limitations and modifications of the reported systems.
Reviewer 2 Report
The authors have collected and summarized the recent developments of nanomaterials and active targeting ligands used for RA gene therapy. Furthermore, the authors have shown various gene delivery systems for future RA treatment. This review can inspire more nanomaterial design ideas in nucleic acid delivery system for rheumatoid arthritis therapy. Overall, this is well-written and well-organized review. Therefore, I would like to recommend this work to publish in Pharmaceutics. Below are some comments for the authors.
1. All examples used in the section “Nanomaterials for RA gene therapy” should be collected as a table. The table could help readers to read this review more easily.
2. The caption of Table 1 should be corrected.
3. In “2. Nanomaterials for RA gene therapy”, for the introduction “Meanwhile, lots of nanomaterials possess specific ligands...”, more references could be cited to broaden the introduction.
https://doi.org/10.1021/acssuschemeng.9b03048
